# Longitudinal SARS-CoV-2 antibody study using the Easy Check COVID-19 IgM/IgG™ lateral flow assay

Renee L. Higgins[1]*, Stephen A. Rawlings[2], Jamie Case[3], Florence Y. Lee[1], Clarence W. Chan[4], Bethany Barrick[3], Zoe C. Burger[2], Kiang-Teck J. Yeo[4], Dena Marrinucci[1]

1 Truvian Sciences, San Diego, California, United States of America, 2 Department of Medicine, Division of Infectious Diseases and Global Public Health, University of California, San Diego, La Jolla, California, United States of America, 3 Scripps Clinic Bio-Repository and Bio-Informatics Core, La Jolla, California, United States of America, 4 Department of Pathology, University of Chicago, Chicago, Illinois, United States of America

* renee.higgins@truvianhealth.com

**Data Availability Statement:** All relevant data are within the manuscript and its Supporting Information files.

## Abstract

Since the initial identification of the novel coronavirus SARS-CoV-2 in December of 2019, researchers have raced to understand its pathogenesis and begun devising vaccine and treatment strategies. An accurate understanding of the body's temporal immune response against SARS-CoV-2 is paramount to successful vaccine development and disease progression monitoring. To provide insight into the antibody response against SARS-CoV-2, plasma samples from 181 PCR-confirmed COVID-19 patients collected at various time-points post-symptom onset (PSO) were tested for the presence of anti-SARS-CoV-2 IgM and IgG antibodies via lateral flow. Additionally, 21 donors were tracked over time to elucidate patient-specific immune responses. We found sustained levels of anti-SARS-CoV-2 antibodies past 130 days PSO, with 99% positivity observed at 31–60 days PSO. By 61–90 days PSO, the percentage of IgM-/IgG+ results were nearly equal to that of IgM+/IgG+ results, demonstrating a shift in the immune response with a decrease in IgM antibody levels. Results from this study not only provide evidence that the antibody response to COVID-19 can persist for over 4 months, but also demonstrates the ability of Easy Check™ to monitor seroconversion and antibody response of patients. Easy Check was sufficiently sensitive to detect antibodies in patient samples as early as 1–4 days PSO with 86% positivity observed at 5–7 days PSO. Further studies are required to determine the longevity and efficacy of anti-SARS-CoV-2 antibodies, and whether they are protective against re-infection.

## Introduction

SARS-CoV-2 is a *betacoronavirus* responsible for causing the inflammatory disease coronavirus disease 2019 (COVID-19) [1–3]. The United States currently leads the world with over 26 million confirmed cases that have resulted in over 443,000 deaths as of February, 2021 [4]. This once-in-a-century pandemic disrupted global economic, education and healthcare systems. Resuming "normal" life in a post-pandemic world requires accurate and rapid diagnostic

**Funding:** The authors received no specific funding for this work.

tests, effective vaccines and efficacious treatments. To this end, a comprehensive understanding of the human immune response against SARS-CoV-2 is essential. Being a novel disease, data is just now emerging with regards to persistence of antibodies following infection [5–7], neutralizing activities of antibodies [7] and T-cell mediated immune response [8, 9].

As the virus infects the host cell, a complex cascade of immune responses involving both B- and T-cells are activated. This results in the generation of virus-specific IgM antibodies within the first week following symptom onset, followed by a longer-lasting IgG antibody response, which could persist for several months or years [10, 11]. Conflicting results have been reported regarding the longevity of the SARS-CoV-2 antibody response [12]. This difference may be attributed to the targets used in the assays employed in the studies. Studies of other coronaviruses have shown a variety of responses, with some antibody-mediated immunity declining as quickly as 12 weeks PSO, while other responses, such as to SARS-CoV and MERS can last from a year to 30 months [5, 6, 13–15]. In contrast, to date there is insufficient data demonstrating indeed how long SARS-CoV-2 immunity can last—unsurprisingly given it has only been infecting humans for less than a year. Further, many of the published studies use loss of detectable antibody interchangeably with loss of immunity. It is important to note that the absence of detectable antibody does not equate to absence of immunity in a patient. Cell-mediated immunity by way of CD4$^+$ or CD8$^+$ T-cells can also be an important indicator of immunity [16–18]. While antibodies are not the sole source of immunity, understanding of the duration and protective effect of this antibody response is critical for informing vaccine strategies and helping to control spread of disease.

There are generally three types of assays (molecular, antigen and serology) commonly used in detecting and thus controlling the spread of infectious diseases. Molecular tests detect the presence of viral genomic material in host samples. These tests can identify the presence of likely active infections. Like molecular tests, antigen tests can detect the presence of active infections as well. Antigen tests differ from molecular tests in that they detect viral proteins present in patient samples. In general, antigen tests tend to enable rapid, point-of-care testing of patients whereas molecular tests tend to be lab-based with longer turnaround times. The third commonly used method is antibody testing also known as serology. Serology tests detect the host's humoral immune response to a viral infection. Unlike molecular and antigen tests, serology tests are not intended to be stand-alone diagnostics for active viral infections but can be leveraged in several other ways. There is growing evidence that serology testing serves as an excellent companion to PCR/antigen testing and can improve detection rates [19–21]. Serology tests are also important for global COVID-19 responses in that they can be utilized for public health response and planning (e.g. sero-epidemiological surveillance), as well as community-based contact tracing [22]. Further, the ability to detect SARS-CoV-2 antibody is useful for convalescent plasma donor identification screening. Additionally, with ongoing SARS-CoV-2 clinical trials, serology tests can confirm seroconversion of patients following vaccination [23].

Serology tests have the potential to become invaluable tools for characterizing the immune response associated with SARS-CoV-2 infection. To date, over 50 serology assays have been granted Emergency Use Authorization (EUA) by the Food and Drug Administration (FDA) in the United States. All serology assays that have gained EUA approval in the US use either the SARS-CoV-2 nucleocapsid only, the spike protein only (either full length or the receptor binding domain [RBD]), or a combination of the two as antigens for detecting antibodies present in the sample. Antibodies against the spike protein, specifically the spike RBD, have been linked to viral neutralizing activity [24, 25]. This correlation between levels of spike RBD-specific antibodies and neutralizing activity suggests that positive results from serological assays that utilize spike RBD antigen may be indicative of a decreased risk of SARS-CoV-2 infection.

Using the nucleocapsid antigen in combination with the spike RBD antigen will help produce more highly sensitive, specific and clinically informative serological assays. Easy Check utilizes both antigens and has been independently validated to be a robust assay that demonstrated a sensitivity of 96.6.%, a specificity of 98.2% and an overall accuracy of 98.1% [26].

In this study, a longitudinal evaluation of the presence of IgM and IgG antibodies using Truvian's Easy Check COVID-19 IgM/IgG™ lateral flow assay was conducted. By evaluating the antibody response using Easy Check, we demonstrate the utility of using a rapid test such as Easy Check in monitoring seroconversion as well as seroprevalence of SARS-CoV-2.

## Materials and methods

### Ethical approvals and study participants

Samples were obtained under study protocols approved by the Institutional Review Board at the University of California, San Diego, or the Scripps Health Institutional Review Board or the University of Chicago. All participants provided written, informed consent. Eligible participants were adults aged ≥18 years who were diagnosed with SARS-CoV-2 by approved nasopharyngeal polymerase chain reaction (PCR) testing conducted by their physician. SARS-CoV-2 PCR tests used includes Abbott ID NOW, Hologic Panther Fusion, ABI7500, ABI7499, CDC 2019-nCoV Real-Time RT-PCR Diagnostic Panel (CDC), Cobas SARS-CoV-2, Quest SARS-CoV-2 rRT-PCR, ePlex SARS-CoV-2 Test. De-identified clinical data including patient demographic information and clinical outcome were retrieved from medical records or via participant interview. The patient population consisted of persons with a range of mild to severe symptoms with 17 known fatalities. Samples were originally collected between March 31st, 2020-August 6th, 2020 and assayed using Easy Check between May 21st, 2020-August 20th, 2020. Details of the patient demographics can be found in Table 1.

**Table 1. Demographics of study participants and sample numbers per participant.**

| Sample Size | |
|---|---|
| Total Patients | 181 |
| Total Samples | 323 |
| **Age** | |
| Median [Min, Max] | 59 [18, 89] |
| IQR | 26 |
| **Sex** | |
| Male | 99 (54.7%) |
| Female | 82 (45.3%) |
| **Draw Timepoints per Participant** | |
| Mean [Min, Max] | 2 [1, 11] |
| IQR | 1 |
| **Samples Included at Each Timepoint** | |
| 1–4 days PSO | 21 |
| 5–7 days PSO | 42 |
| 8–30 days PSO | 147 |
| 31–60 days PSO | 69 |
| 61–90 days PSO | 24 |
| 91–144 days PSO | 20 |

Summary of the 181 participants and 323 samples from this study. Samples were divided into cohorts based on days post symptom onset (PSO).

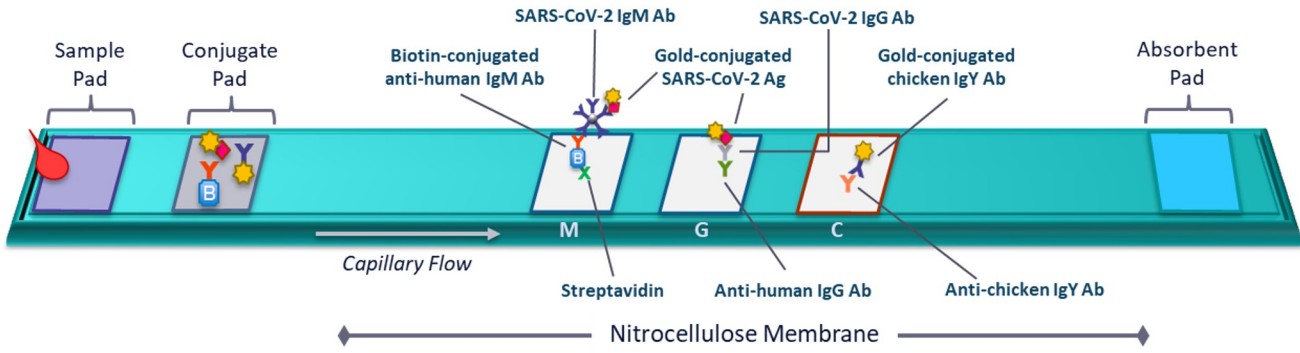

**Fig 1. Schematic of Easy Check COVID-19 IgM/IgG<sup>TM</sup> lateral flow assay.**

## Blood sample processing and storage

Participant blood was collected via venipuncture into K2-EDTA or citrate tubes. Plasma was isolated by centrifugation, stored at 4˚C for up to 7 days before use or frozen at -80˚C for long-term storage. Prior to experiments, aliquots of plasma samples were warmed to room temperature (25–30˚C).

## Lateral flow assay

As previously described by Chan et al., the Easy Check COVID-19 IgM/IgG™ test is an immunochromatographic assay for the detection of SARS-CoV-2 IgM and/or IgG antibodies in human plasma specimens [26]. The test uses the spike RBD and the nucleocapsid protein as antigens. Control antibody, anti-human IgG and streptavidin (test line for IgM) are immobilized onto a nitrocellulose membrane to form three distinct lines: the control line, IgG test line and IgM test line (Fig 1). The nitrocellulose membrane is attached onto a plastic backing card and combined with other reagents and pads to construct a test strip. The test strip is encased inside a plastic device (Fig 2).

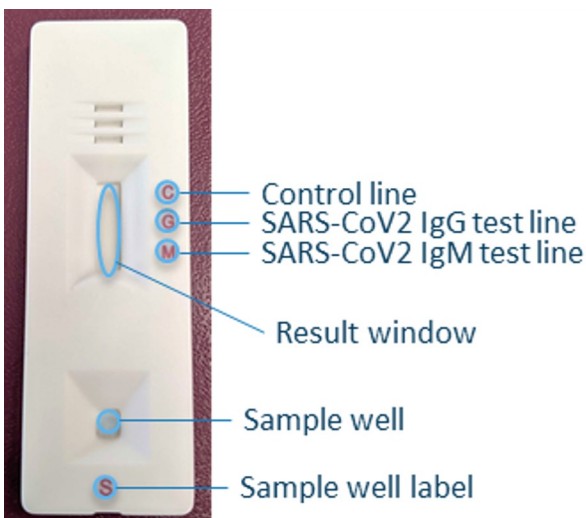

**Fig 2. Easy Check COVID-19 IgM/IgG<sup>TM</sup> lateral flow test strip depicting the various components of Easy Check.**
Sample is loaded into sample well and wicks into test strip. Lines can be visualized in the result window. A band will appear at C (control line) if the test is valid. A line will appear at the SARS-CoV-2 IgG (G) test line if the sample is positive for anti-SARS-CoV-2 IgG antibodies. A line will appear at the SARS-CoV-2 IgM (M) test line if the sample is positive for anti-SARS-CoV-2 IgM antibodies.

Sample is loaded into the sample port and wicks into the test strip via capillary flow. The sample specimens migrate sequentially through filter pad, conjugate pad, nitrocellulose membrane, and absorbent pad. Present on the conjugation pad are SARS-CoV-2 antigens conjugated to gold nanoparticles, gold conjugated chicken IgY antibody, and biotin conjugated anti-human IgM antibodies. Anti-SARS-CoV-2 IgM antibodies present in the sample will bind to the biotin conjugated anti-human IgM antibodies and gold conjugated antigens at the conjugate pad before binding to the streptavidin present at the "M" test line on the nitrocellulose membrane. Anti-SARS-CoV-2 IgG antibodies present in the sample will bind to the gold conjugated antigens present at the conjugate pad before migrating further through the strip and binding to the anti-human IgG antibody at the "G" test line on the nitrocellulose membrane. Sample flowing through the test strip will carry gold conjugated chicken IgY antibodies present at the conjugate pad through the test strip where they will bind to the anti-chicken IgY antibody located at the "C" line on the nitrocellulose membrane.

The test results were visually interpreted 10 minutes after starting the test. Any colored lines in the test region were considered as positive regardless of line intensity. The presence of two lines marked by "C" and "G" indicates SARS-CoV-2 IgG positivity. The presence of two lines marked by "C" and "M" indicates SARS-CoV-2 IgM positivity. The presence of three lines "C", "G" and "M" indicates positivity for both SARS-CoV-2 IgG and IgM. The appearance of only the control line "C" indicates a correctly performed test, but also that the sample is negative for SARS-CoV-2 IgM and IgG.

## Results

### Demographics of SARS-CoV-2 infected patients

A total of 323 samples acquired from 181 participants were used in this study. An average of 2 collection times (interquartile range or IQR = 1) per participant were used in this study. The median participant age was 59 years old (IQR = 26). 55% of study participants were male and 45% of participants were female. Samples were divided into groups based on the number of days PSO. Each group consisted of at least 20 samples (Table 1).

### Percent positive over time

We evaluated percent positivity of samples over time from one day PSO through 144 days PSO. Easy Check was able to detect positive samples as early as 1–4 days PSO with a percent positivity of 52%. IgM showed its highest level of positivity (90%) among samples at 8–30 days PSO (Table 2). In contrast, IgG showed its highest level of positivity (99%) among samples at

**Table 2. IgM and IgG percent positive over time.**

| | IgM | | | | IgG | | | | IgM and/or IgG | | | |
|---|---|---|---|---|---|---|---|---|---|---|---|---|
| Days PSO | N | Positive | % | 95% CI | N | Positive | % | 95% CI | N | Positive | % | 95% CI |
| 1–4 | 21 | 11 | 52% | 32.4–71.7 | 21 | 7 | 33% | 17.2–54.6 | 21 | 11 | 52% | 32.4–71.7 |
| 5–7 | 42 | 35 | 83% | 69.4–91.7 | 42 | 30 | 71% | 56.4–82.8 | 42 | 36 | 86% | 72.2–93.3 |
| 8–30 | 147 | 133 | 90% | 84.6–94.2 | 147 | 140 | 95% | 90.5–97.7 | 147 | 143 | 97% | 93.2–98.9 |
| 31–60 | 69 | 59 | 86% | 75.3–91.9 | 69 | 68 | 99% | 92.2–99.9 | 69 | 68 | 99% | 92.2–99.9 |
| 61–90 | 24 | 12 | 50% | 31.4–68.6 | 24 | 23 | 96% | 79.8–99.8 | 24 | 23 | 96% | 79.8–99.8 |
| 91–144 | 20 | 9 | 45% | 31.0–73.8 | 20 | 19 | 95% | 73–99.7 | 20 | 19 | 95% | 73–99.7 |

323 samples from 181 patients were split into cohorts based on days post symptom onset (PSO). Percent positivity for IgM and IgG was calculated at each time cohort. 95% confidence interval calculated using Wilson/Brown method. N = total samples.

31–60 days PSO. While percent positivity of IgM declined over time, 45% of samples had IgM detectable by Easy Check past 90 days PSO. Notably, IgG percent positivity persisted at greater than 90% beyond 90 days PSO (Table 2).

### Antibodies at various days post onset of symptoms

IgM was detectable early in the COVID-19 disease course. At 5–7 days PSO, 14% of samples were positive for IgM only. During this same time frame, 69% of samples were positive for both IgM and IgG (Fig 3). This fraction of samples testing positive for both IgM and IgG increased to 88% between 8–30 days PSO. 7% of samples were negative for IgM and positive for IgG, whereas 3% of samples were negative for both IgM and IgG. Between 31–60 days PSO, there were no samples that were IgM+/IgG-; additionally, only 1% of samples tested were negative for antibodies (Fig 3). Between 91–144 days PSO, the negative fraction increased to 5% while the IgM+/IgG+ and IgM-/IgG+ fractions decreased to 45% and 50%, respectively (Fig 3). Overall, these results demonstrate a trend of IgM rising early in the recovery phase and then waning in the later stages, when IgG predominates. Of note, we did not observe a change in the number of negative samples from 31 to 144 days PSO. This finding is consistent with previously reported studies that found SARS-CoV-2 antibodies persisting for at least 3 months PSO [7], and in disagreement with a number of studies that have reported short lived immunity observed in COVID-19 patients [5, 27].

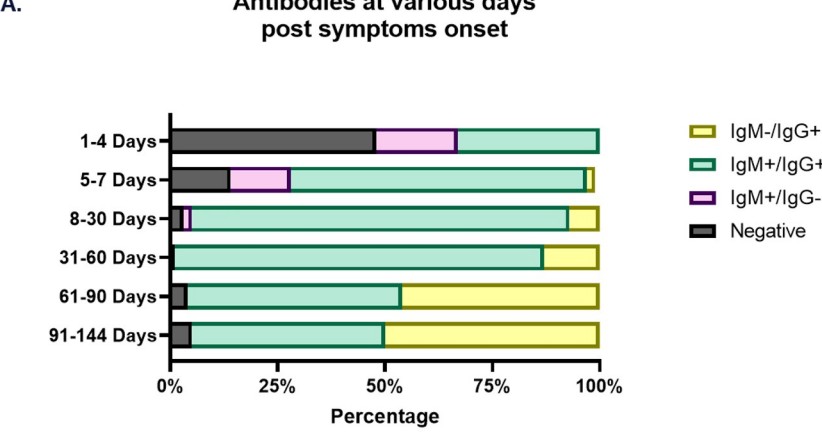

| | IgM+/IgG+ | IgM-/IgG+ | IgM+/IgG- | Negative | Positive | N |
|---|---|---|---|---|---|---|
| **1-4 Days** | 33% (7) | 0% (0) | 19% (4) | 48% (10) | 52% (11) | 21 |
| **5-7 Days** | 69% (29) | 2% (1) | 14% (6) | 14% (6) | 86% (36) | 42 |
| **8-30 Days** | 88% (130) | 7% (10) | 2% (3) | 3% (4) | 97% (143) | 147 |
| **31-60 Days** | 86% (59) | 13% (9) | 0% (0) | 1% (1) | 99% (68) | 69 |
| **61-90 Days** | 50% (12) | 46% (11) | 0% (0) | 4% (1) | 96% (23) | 24 |
| **91-144 Days** | 45% (9) | 50% (10) | 0% (0) | 5% (1) | 95% (19) | 20 |

**Fig 3. Antibody response at various days post symptom onset.** 323 samples from 181 patients were split into cohorts based on days post-symptom onset. A) Bar graph depicts percentage of patient samples at each time range that are negative (grey bar), positive for IgG only (yellow bar), positive for IgM only (lavender bar) or positive for both IgM and IgG (green bar). B) Corresponding table indicating percentage of samples that are positive for both IgM and IgG, positive for IgG only or positive for IgM only along with the total percent negative and positive at each time cohort. Total number of samples in each category: IgM+/IgG+, IgM-/IgG+, IgM+/IgG-, Negative, Positive at each time range indicated in parentheses. N = total samples at each time cohort.

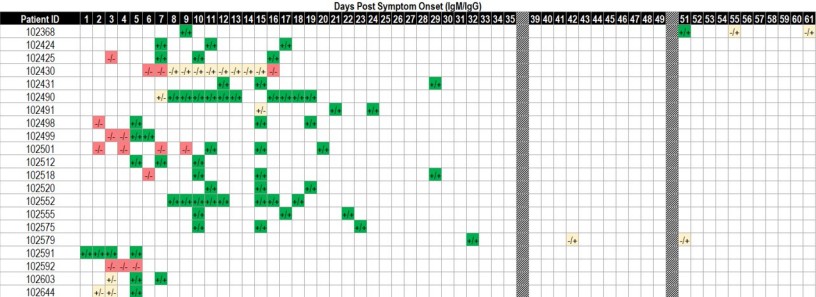

**Fig 4. Individual longitudinal antibody response.** Antibody responses for patients who had 3 or more samples collected over the course of the study. This subset of patient samples was collected 1–61 days post-symptom onset. Results are depicted as IgM result/IgG result where red indicates negative result, yellow indicates positive for IgM or IgG, dark green indicates positive for both IgM and IgG.

## Individual participant trends observations

Twenty-one participants had 3 or more repeat sample draws over the course of 2 months. Of those participants, 4 showed a trend of antibodies appearing between 1–5 days PSO. Nineteen of 20 participants who were tested between 1–20 days PSO showed the presence of antibodies by day 15. Participant 102430 was a case with an interesting antibody response. This participant was one of only two out of the group of 21 that did not develop detectable IgM antibodies. This participant had no detectable IgM levels from 6–16 days PSO, and detectable IgG that persisted to day 15 PSO before becoming undetectable (Fig 4). Furthermore, this particular participant received convalescent plasma treatment 7 days PSO. It is therefore possible that the detected IgG was due to the convalescent plasma treatment and not from the participant's own immune response. The other participant who had no detectable IgM levels, 102592, was tested on days 3, 4 and 5 PSO and was negative for both IgM and IgG. It is possible that this participant developed antibodies after the day 5 timepoint.

## Discussion

Here, we used the Easy Check COVID-19 IgM/IgG test to measure the presence of anti-SARS-CoV-2 antibodies in 323 samples from 181 confirmed COVID-19 positive participants, and additionally with longitudinal time points for several study participants. The goal of this study was to address the current knowledge deficit on the duration of the antibody response to SARS-CoV-2, as well as to demonstrate the utility of using lateral flow assays, specifically, Easy Check for longitudinal and seroprevalence studies. Conflicting studies have reported varying duration of antibody response to SARS-CoV-2, with one study showing declining antibodies levels as early as 2 months PSO [12, 27]. These findings have raised alarms that the SARS-CoV-2 immune response in general may be short-lived, causing concerns about the durability of vaccine-induced protection. Notably, our study demonstrated that 95% of samples tested positive for IgG up to 91–144 days PSO. While the number of patients followed over 90 days is limited, our finding is in line with other recent studies showing antibodies persisting for at least 3 months post symptom onset [28, 29]. Additionally, our data showed that IgG antibodies can be detected at least 144 days PSO, with the majority of samples being IgM-/IgG+ or IgM +/IgG+ after two months PSO. Samples positive for IgM alone were identified from 5 days through one-month PSO. Our finding also supports the utility of a SARS-CoV-2 vaccine, since the humoral immune response appears to persist and is likely protective against reinfection. Future studies are currently ongoing to explore how antibodies persist at 8 months and beyond.

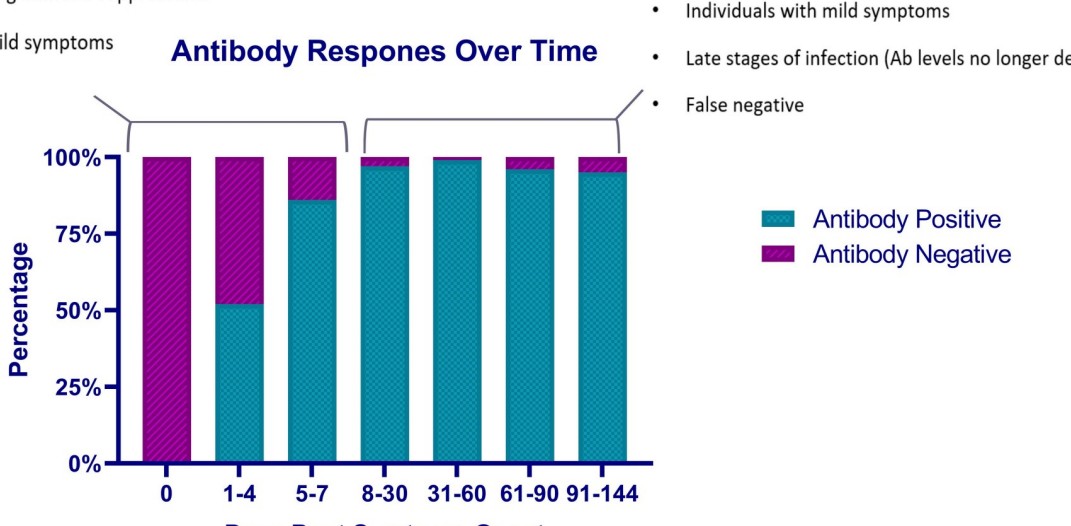

**Potential Causes For Seronegative Individuals (0-7 days PSO)**

- Early stages of infection (Ab production has not begun)
- Immunocompromised individuals
- Individuals receiving immune suppressants
- Individuals with mild symptoms
- False negative

**Potential Causes For Seronegative Individuals (8-144 days PSO)**

- Early stages of infection (Ab production has not begun)
- Immunocompromised individuals
- Individuals receiving immune suppressants
- Individuals with mild symptoms
- Late stages of infection (Ab levels no longer detectable)
- False negative

**Fig 5. Potential causes for seronegative individuals across days post-symptom onset.** Graphical representation of percent positivity over time based off of 323 samples from 181 individuals. Pink bars highlight seronegative samples ranging from 100% presumed negative at day 0 to ~5% negative rate at 91–144 days PSO. Blue bars represent seropositive samples.

Seronegative samples were detected throughout the PSO time range investigated with the highest percent negative occurring between 1–7 days PSO (Figs 3 and 5). A small fraction of negative samples was detected between days 31–144 PSO. There are several potential causes for these seronegative samples (Fig 5). For the seronegative samples occurring early in infection, it is possible that the individual has not yet had time to fully mount an immune response that is detectable by the Easy Check assay. This was observed in our individual longitudinal investigation where some individuals did not begin to seroconvert until days 5–8 PSO (Fig 4). Conversely, seronegativity occurring later in infection could be due to an individual being in the late stages of infection and, as a result, they have antibody levels that are below the level of detection for the Easy Check assay. The potential general causes for seronegative samples regardless of days PSO include individuals who are immunocompromised or on immunosuppressants, individuals with mild symptoms who produced only low levels of antibodies in response to COVID-19 or false negative samples (Fig 4) [30–32]. Easy check was independently validated with a determined sensitivity of 96.6% and specificity of 98.2% [26]. Because sensitivity and specificity are not 100%, it is possible that a number of the seronegative samples are false negatives or that some of the seropositive samples are false positives.

It is important to note that antibodies account only partially for immunity against viral and bacterial pathogens. Cell-mediated immunity is also a key contributor in the fight against such infections, including influenza viruses, the rabies virus, *Mycobacterium tuberculosis*, and *Listeria* [33]. Several recent studies demonstrated that T-cell (CD4[+] and CD8[+]) mediated immunity has a critical role in SARS-CoV-2 infections [8, 9, 16–18]. Furthermore, declining

antibody concentration does not necessarily equate to a loss of immunity. A notable example is the antibody response induced by the smallpox vaccine, which decreases by roughly 75% after 6 months while immunity lasts for decades [34]. Low levels of antibodies produced by memory B cells could be sufficient to mount an effective immune response upon re-infection. Studies of other coronaviruses have shown a variety of responses, with some antibody-mediated immunity declining as quickly as 12 weeks PSO while others such as SARS-CoV and MERS can last from a year to 17 years [5, 6, 8, 13–15]. Studies in non-human primates also confirmed the ability to induce antibody response that is protective against re-infection [35–37].

A large number of SARS-CoV-2 infections are asymptomatic, with one study finding at least 6 times more infections per site by seroprevalence assays than with antigen or molecular tests [38]. In Spain, a study found that 23–36% of seropositive patients were asymptomatic [39]. Such findings highlight the importance of serology tests to bridge the data gap and gain a full picture of disease impact. In summary, our findings show the utility of the Easy Check COVID-19 IgM/IgG test as early as 5 days PSO as well as the ability to detect antibody at $\geq 4$ months. This persistence of antibodies past 4 months PSO indicate the likelihood of reinfection within the first 5 months following initial infection with SARS-CoV-2 is low. This is further supported by the current relatively low number of confirmed re-infections [40].

Rapid, point-of-care serology tests are critical in combating the current pandemic. While serology tests are not designed to diagnose active infections, these low cost, highly deployable and easy-to-use tests provide major advantages in several contexts, including 1) routine seroprevalence monitoring in specific sentinel sites to provide information of virus circulation and inform community-based contact tracing [22], 2) confirmation of prior infection for individuals who were unable to receive a molecular test, 3) detection of infections in asymptomatic individuals through serosurveys, 4) confirmation of seroconversion following vaccination, 5) identification of convalescent plasma donors, and 6) seroprevalence in special populations such as the meat packing industry or assisted living facilities, to compare infection rates to the larger community and help identify mitigation measures to prevent the spread of disease within these communities.

## Supporting information

**S1 Dataset.**
(XLSX)

## Acknowledgments

We thank the Scripps Clinic Bio-Repository for providing some of the samples used in this study. We thank the Davey Smith lab at UCSD for providing some of the samples used in this study.

## Author Contributions

**Conceptualization:** Renee L. Higgins, Florence Y. Lee, Dena Marrinucci.

**Data curation:** Renee L. Higgins, Stephen A. Rawlings, Jamie Case, Florence Y. Lee, Clarence W. Chan, Bethany Barrick, Zoe C. Burger, Kiang-Teck J. Yeo, Dena Marrinucci.

**Formal analysis:** Renee L. Higgins, Kiang-Teck J. Yeo, Dena Marrinucci.

**Investigation:** Renee L. Higgins.

**Project administration:** Zoe C. Burger.

**Writing – original draft:** Renee L. Higgins, Stephen A. Rawlings.

**Writing – review & editing:** Renee L. Higgins, Stephen A. Rawlings, Jamie Case, Florence Y. Lee, Clarence W. Chan, Bethany Barrick, Kiang-Teck J. Yeo, Dena Marrinucci.

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
