## [Decision Letter · Decision Letter 0]

11 Jan 2021

PONE-D-20-34194

Longitudinal SARS-CoV-2 antibody study using the Easy Check COVID-19 IgM/IgGTM lateral flow assay

PLOS ONE

Dear Dr. Higgins,

Thank you for submitting your manuscript to PLOS ONE. After careful consideration, we feel that it has merit but does not fully meet PLOS ONE’s publication criteria as it currently stands. Therefore, we invite you to submit a revised version of the manuscript that addresses the points raised during the review process.

Your manuscript was reviewed by one expert in the field. Other potential reviewers did not accept the invitation to review your paper. The reviewer identified several important problems in your submission. Especially, please note the comment on comparison to a gold standard. Please consider the attached comments and provide your point-by-point responses.    

We look forward to receiving your revised manuscript.

Kind regards,

Yury E Khudyakov, PhD

Academic Editor

PLOS ONE

Journal Requirements:

2. Please include the date range on which samples were originally collected as well as the date(s) you obtained the retrospective data.

"This research was supported in part by the San Diego Center for AIDS Research (SD CFAR) Translational Virology Core, an NIH-funded program (P30 AI036214)."

"The authors received no specific funding for this work."

"I have read the journal's policy and the authors of this manuscript have the following competing interests: Renee Higgins, Florence Lee and Dena Marrinucci are employees of Truvian Sciences, the company that sells Easy Check. Kiang-Teck J. Yeo is a member of the Scientific Advisory Board of Truvian and has received honoraria and equities from Truvian."

5. We note that Figures 1 and 2 in your submission contain copyrighted images. All PLOS content is published under the Creative Commons Attribution License (CC BY 4.0), which means that the manuscript, images, and Supporting Information files will be freely available online, and any third party is permitted to access, download, copy, distribute, and use these materials in any way, even commercially, with proper attribution. For more information, see our copyright guidelines: http://journals.plos.org/plosone/s/licenses-and-copyright.

(1) You may seek permission from the original copyright holder of Figures 1 and 2 to publish the content specifically under the CC BY 4.0 license.

6. We note that you have indicated that data from this study are available upon request. PLOS only allows data to be available upon request if there are legal or ethical restrictions on sharing data publicly. For information on unacceptable data access restrictions, please see http://journals.plos.org/plosone/s/data-availability#loc-unacceptable-data-access-restrictions.

Reviewers' comments:

Reviewer's Responses to Questions

**Comments to the Author**

1. Is the manuscript technically sound, and do the data support the conclusions?

Reviewer #1: Yes

2. Has the statistical analysis been performed appropriately and rigorously? 

Reviewer #1: Yes

3. Have the authors made all data underlying the findings in their manuscript fully available?

Reviewer #1: Yes

4. Is the manuscript presented in an intelligible fashion and written in standard English?

Reviewer #1: Yes

5. Review Comments to the Author

Reviewer #1: the paper appears to add some valuable new information regarding a rapid test for antibody positivity to SARS-CoV2 as well as some information about duration of IgM and IgG antibodies. In general it is well written. there a some critiques as follows:

1. The assay is not compared to another gold standard assay within this study. It was validated in another study but the authors should at least discuss the potential for false positives and false negatives within their data set based on prior validation

2. The gold standard used to determine positivity for COVID 19 should be provided

3. There is little data on the clinical course of the patients - was any data available?

4. The authors should discuss what their findings add to the existing literature - Conclusions are drawn about the duration of antibody positivity but the number of patients followed for over 90 days is limited

5. The final figure is very hard to interpret

6. PLOS authors have the option to publish the peer review history of their article (what does this mean?). If published, this will include your full peer review and any attached files.

Reviewer #1: **Yes: **Kevan Hartshorn

---

## [Author Response · Author response to Decision Letter 0]

10 Feb 2021

Dear Editors, 

Thank you for your feedback regarding our manuscript, Longitudinal SARS-CoV-2 Antibody Study Using the Easy Check COVID-19 IgM/IgG Lateral Flow Assay. You have made very good points regarding several aspects of our paper. We will address each point below. 

Author Response: We have revised our manuscript to ensure that it meets PLOS ONE’s style requirements, including those for file naming. 

 2. Please include the date range on which samples were originally collected as well as the date(s) you obtained the retrospective data.

Author Response: We have added the date range on which samples were originally collected as well as the dates we obtained the retrospective data. 

"This research was supported in part by the San Diego Center for AIDS Research (SD CFAR) Translational Virology Core, an NIH-funded program (P30 AI036214)."

"The authors received no specific funding for this work."

Author Response: We apologize for the confusion. One of our authors, Stephen Rawlings salary is covered by the SD CFAR Tranlsational Virology Core, an NIH-funded program. This is not specific funding for our project but rather a general funding source such that Stephen can execute research related to SD CFAR Translational Virology Core initiatives. Our funding statement remains that we received no specific funding for this work. We will remove funding-related text from the manuscript. 

"I have read the journal's policy and the authors of this manuscript have the following competing interests: Renee Higgins, Florence Lee and Dena Marrinucci are employees of Truvian Sciences, the company that sells Easy Check. Kiang-Teck J. Yeo is a member of the Scientific Advisory Board of Truvian and has received honoraria and equities from Truvian."

Author Response: Updated Competing Interests statement is as follows: I have read the journal's policy and the authors of this manuscript have the following competing interests: Renee Higgins, Florence Lee and Dena Marrinucci are employees of Truvian Sciences, the company that sells Easy Check. Kiang-Teck J. Yeo is a member of the Scientific Advisory Board of Truvian and has received honoraria and equities from Truvian. This does not alter our adherence to PLOS ONE policies on sharing data and materials.

5. We note that Figures 1 and 2 in your submission contain copyrighted images. All PLOS content is published under the Creative Commons Attribution License (CC BY 4.0), which means that the manuscript, images, and Supporting Information files will be freely available online, and any third party is permitted to access, download, copy, distribute, and use these materials in any way, even commercially, with proper attribution. For more information, see our copyright guidelines: http://journals.plos.org/plosone/s/licenses-and-copyright.

(1) You may seek permission from the original copyright holder of Figures 1 and 2 to publish the content specifically under the CC BY 4.0 license.

Author Response: We have received permission for PLOS One to publish Figures 1 and 2 from AJCP. We have uploaded proof of granted permissions as an “Other” file with our submission. 

6. We note that you have indicated that data from this study are available upon request. PLOS only allows data to be available upon request if there are legal or ethical restrictions on sharing data publicly. For information on unacceptable data access restrictions, please see http://journals.plos.org/plosone/s/data-availability#loc-unacceptable-data-access-restrictions.

Author Response: There are no legal or ethical restrictions on sharing our data publicly. We have uploaded our minimal anonymized data set as supporting information files. 

Reviewer #1: the paper appears to add some valuable new information regarding a rapid test for antibody positivity to SARS-CoV2 as well as some information about duration of IgM and IgG antibodies. In general it is well written. there are some critiques as follows:

1. The assay is not compared to another gold standard assay within this study. It was validated in another study but the authors should at least discuss the potential for false positives and false negatives within their data set based on prior validation

Author Response: Thank you for your feedback. This is a valid point. We have added a statement to our manuscript to highlight the possibility of false positives and false negatives within our data set. 

2. The gold standard used to determine positivity for COVID 19 should be provided

Author Response: There were multiple gold standard PCR tests used across the 3 sites. We have updated our methods section to include a list of all PCR tests used.

3. There is little data on the clinical course of the patients - was any data available?

Author Response: Complete data on subject symptom severity were not available for all samples. However, within cohorts that had symptom severity, the majority of participants had mild and moderate symptoms, and very few had severe symptoms—similar to the distribution of clinical outcomes for the pandemic as a whole. We do not have complete follow-up data on all participants either but are only aware of 17 fatalities.

4. The authors should discuss what their findings add to the existing literature - Conclusions are drawn about the duration of antibody positivity but the number of patients followed for over 90 days is limited

Author Response: Thank you for your feedback. We have added text to our manuscript to highlight what our findings add to the existing literature as well as discussing the limitations of the findings. 

5. The final figure is very hard to interpret

Author Response: We have updated figure 5 to make the message clearer. We’re open to additional feedback if it is still difficult to follow. 

We look forward to hearing your decision on our manuscript. Thank you for your time reviewing our submission and response. 

Kind regards, 

Reneé Higgins, PhD

Manager, Scientific Affairs 

Truvian

10300 Campus Point Dr

Suite 190

San Diego, CA 92121

---

## [Editor Report · Decision Letter 1]

15 Feb 2021

Longitudinal SARS-CoV-2 antibody study using the Easy Check COVID-19 IgM/IgGTM lateral flow assay

PONE-D-20-34194R1

Dear Dr. Higgins,

We’re pleased to inform you that your manuscript has been judged scientifically suitable for publication and will be formally accepted for publication once it meets all outstanding technical requirements.

Kind regards,

Yury E Khudyakov, PhD

Academic Editor

PLOS ONE
---

## [Editor Report · Acceptance letter]

23 Feb 2021

PONE-D-20-34194R1 

Longitudinal SARS-CoV-2 Antibody Study Using the Easy Check COVID-19 IgM/IgG^™^ Lateral Flow Assay 

Dear Dr. Higgins:

I'm pleased to inform you that your manuscript has been deemed suitable for publication in PLOS ONE. Congratulations! Your manuscript is now with our production department. 

Kind regards, 

on behalf of

Dr. Yury E Khudyakov 

Academic Editor

PLOS ONE